# Transferrin-Targeted Liposomes in Glioblastoma Therapy: A Review

**DOI:** 10.3390/ijms241713262

**Published:** 2023-08-26

**Authors:** Paul Kawak, Nour M. Al Sawaftah, William G. Pitt, Ghaleb A. Husseini

**Affiliations:** 1Chemical and Biological Engineering Department, College of Engineering, American University of Sharjah, Sharjah P.O. Box 26666, United Arab Emirates; paulkawak@gmail.com; 2Materials Science and Engineering Program, College of Arts and Sciences, American University of Sharjah, Sharjah P.O. Box 26666, United Arab Emirates; nalsawaftah@aus.edu; 3Chemical Engineering Department, Brigham Young University, Provo, UT 84602, USA

**Keywords:** cancer, glioblastoma, blood-brain barrier, liposomes, targeted drug delivery, transferrin

## Abstract

Glioblastoma (GBM) is a highly aggressive brain tumor, and its treatment is further complicated by the high selectivity of the blood–brain barrier (BBB). The scientific community is urgently seeking innovative and effective therapeutic solutions. Liposomes are a promising new tool that has shown potential in addressing the limitations of chemotherapy, such as poor bioavailability and toxicity to healthy cells. However, passive targeting strategies based solely on the physicochemical properties of liposomes have proven ineffective due to a lack of tissue specificity. Accordingly, the upregulation of transferrin receptors (TfRs) in brain tissue has led to the development of TfR-targeted anticancer therapeutics. Currently, one of the most widely adopted methods for improving drug delivery in the treatment of GBM and other neurological disorders is the utilization of active targeting strategies that specifically target this receptor. In this review, we discuss the role of Tf-conjugated liposomes in GBM therapy and present some recent studies investigating the drug delivery efficiency of Tf-liposomes; in addition, we address some challenges currently facing this approach to treatment and present some potential improvement possibilities.

## 1. Introduction

In normal, healthy cells, the process of cell growth and division is intricately controlled by hundreds of genes that promote proliferation, suppress it, and/or signal when cells should undergo apoptosis. Cancer cells are genetically mutated, rendering them unresponsive to many signals that control cellular growth and death (refer to Figure 1) [1,2]. 

Chemotherapy is a frequently employed treatment modality for cancer. It employs cytotoxic drugs which interfere with DNA synthesis and cell division to kill fast-growing cells [3]. However, chemotherapeutic agents are nonselective and kill fast-growing cells indiscriminately; hence, chemotherapy is associated with detrimental systemic side effects such as hair loss, nausea, vomiting, fatigue, and mouth sores [4,5,6]. The emergence of nanotechnology has profoundly impacted clinical therapeutics, particularly in the field of cancer therapy. Compared to conventional chemotherapy, nanoscale drug carriers can reduce the toxicity towards healthy cells by concealing the drug while it circulates in the body and releasing it at the target sites; in addition, nanocarriers can improve treatment efficacy by increasing the drug concentration at the target site without the need for increasing the dosage [7,8,9,10]. Several nanocarriers have been investigated as drug-delivery vehicles for anticancer agents, such as dendrimers, solid lipid nanoparticles, hydrogels, micelles, metal-organic frameworks, and liposomes. Among these emergent nanocarriers, liposomes have been used most for clinical applications [11,12].

### 1.1. Liposomes

Liposomes are nanosized to microsized synthetic spherical vesicles made up of a phospholipid bilayer surrounding an aqueous spherical core, similar in structure to that found in cells [3,13]. Each phospholipid molecule is composed of a hydrophilic head and a hydrophobic tail, which, when dispersed in an aqueous medium, arrange themselves such that the hydrophilic heads of phospholipids are directed outward, facing the aqueous environment, whereas the hydrophobic tails are directed inwards (see Figure 2). This arrangement gives rise to the amphiphilic nature of liposomes which can accommodate hydrophilic substances in their aqueous core, and hydrophobic substances in the hydrophobic tail region [14,15,16]. Depending on the synthesis method, the sizes of liposomes can vary from tens of nanometers to several micrometers. Due to their biocompatibility and low toxicity, extensive research has been conducted on liposomes for their potential in drug delivery applications. They can protect drugs from kidney clearance and degradation in the body, thus enhancing their bioavailability, leading to improved therapeutic outcomes [17,18,19]. At present, several liposome-based chemotherapeutics have been clinically approved and many more are under various stages of clinical or preclinical development (see Table 1).

Moreover, liposomes can be used to deliver drugs to specific cells or tissues as they can be targeted to bind to selected cells by modifying their surface with ligands that bind to specific receptors on the target cells, a technique known as active targeting. A wide range of molecules can be used to functionalize the surfaces of liposomes, including carbohydrates, proteins, peptides, aptamers, monoclonal antibodies (mAbs), and more [8,22,23,24,25]. Actively targeted liposomes can recognize and bind to the target cells through ligand-receptor interactions. After the bound liposomes are internalized (endocytosed), they release their payload inside the cell, resulting in reduced off-target effects compared to passively targeted systems. This is particularly significant in cancer therapy because certain receptors tend to be overexpressed on some cancer cell lines [6,26,27,28]. One of the overexpressed receptors in GBM is the transferrin receptor (TfR), which is the focus of this review [29]. 

While extensive research has explored various aspects of GBM-targeted therapy, the majority of the existing literature focuses on targeting the TfR with a wide array of ligands. Building upon these reviews [30,31,32], this article highlights the potential of utilizing Tf as a ligand, since these liposomes can specifically bind to and enter cancer cells, thereby improving drug accumulation within the tumor site while minimizing off-target effects. This novel strategy holds immense potential for improving the efficacy and precision of GBM treatment, opening up new avenues for personalized and targeted therapies in the battle against this aggressive, malignant disease. Additionally, this review explores cutting-edge research in the field, shedding light on novel studies. This review paper aims to consolidate the existing knowledge while addressing the critical areas that remain unexplored, thereby paving the way for further advancements in GBM therapy utilizing Tf-modified liposomes.

### 1.2. Transferrin and the Transferrin Receptor

Transferrin (Tf) is an 80 kDa glycoprotein found in the blood that plays a crucial role in the transportation of iron throughout the body. It is primarily synthesized by the liver and secreted into the bloodstream. Tf binds to and carries iron ions from areas of absorption, such as the intestines, to areas of utilization, such as the bone marrow, where iron is necessary for the production of new red blood cells [33,34]. The unbound form of Tf is known as apo transferrin (apo-Tf), whereas iron-bound Tf is referred to as holo-Tf. Each Tf molecule can shuttle two iron (Fe^+3^) ions by binding to TfRs on the cell surface; the Fe^+3^/Tf/TfR complex then enters the cells through clathrin-mediated endocytosis [35,36,37]. 

Two types of Tf receptors exist, labeled 1 and 2 (TfR1 and TfR2, respectively). TfR1, also known as CD71, is the major TfR isoform and is widely expressed in many cell types. It plays a crucial role in the uptake of Tf-bound iron into cells. TfR1 is responsible for the endocytosis of Tf and its subsequent internalization into the cell, along with bound iron. It is highly expressed in cells with a high demand for iron, such as developing red blood cells, actively dividing cells, and liver and bone marrow cells. TfR2, the second TfR isoform, shares some structural similarities with TfR1 [30,38,39]. However, TfR2 has a distinct role and tissue distribution compared to TfR1. It is predominantly expressed in hepatocytes (liver cells) and the small intestine and is involved in regulating iron homeostasis by modulating the production of the hormone hepcidin, which regulates iron absorption and distribution. TfR2 has a 25-fold reduced holo-Tf affinity compared to TfR1 and is believed to participate in the overall iron metabolism in addition to the cellular iron uptake [40,41].

When the Tf ligand binds to either of the TfRs, the complex is internalized through clathrin-mediated endocytosis (CME), which is one of the major pathways for receptor-mediated endocytosis and involves the formation of specialized vesicles called clathrin-coated vesicles (CCVs). These vesicles are formed at specific plasma membrane regions known as clathrin-coated pits (CCPs) [38,42]. Clathrin-mediated endocytosis is a highly regulated process, and defects in this pathway can lead to various disease states. It plays a crucial role in nutrient uptake, receptor internalization, synaptic vesicle recycling in neurons, and many other cellular processes. The process of CME can be divided into several steps (refer to Figure 3) [38,43,44]:

Initiation: CME is triggered by the binding of ligands, such as proteins, growth factors, aptamers, hormones, etc., to specific cell surface receptors. This interaction induces receptor clustering and the recruitment of signal-transducing adaptor proteins to the cytoplasmic domain of the receptors.

Assembly of the clathrin coat: Adaptor proteins, such as AP2 (adaptor protein 2), bind to the cytoplasmic tails of the clustered receptors and recruit clathrin molecules to the CCPs. Clathrin is a triskelion-shaped protein that assembles into a lattice structure around the CCPs, providing structural support and driving vesicle formation.

Formation of the clathrin-coated vesicle: the forming vesicle is constricted and eventually pinches off, forming a clathrin-coated vesicle containing the cargo molecules from the extracellular space.

Vesicle fusion and cargo sorting: The clathrin-coated vesicles fuse with early endosomes, and the cargo molecules are delivered to their respective destinations. As cargo molecules enter early endosomes, they are either directed to recycling pathways back to the plasma membrane or sent to late endosomes and lysosomes for degradation.

### 1.3. Conjugation Chemistry of Tf onto Liposomes

The attachment of Tf to a drug-loaded liposome surface and the binding of the Tf to a TfR induces the uptake of the liposomes into a cell. There are several methods to conjugate ligands onto liposomes, depending on the nature of the ligand and the desired conjugation strategy. The conjugation between the ligand and the phospholipid membrane occurs either directly or through a spacer molecule. Below are some commonly used methods of ligand attachment [35,42,43,44,45]:

Conjugation of the targeting moiety prior to the formation of liposomes: this approach involves the synthesis and purification of the lipid-moiety pair prior to the preparation of the liposome. 

Post-insertion method: In this method, liposomes are prepared separately and then the targeting ligands are conjugated onto the surface of the liposomes after their formation. Ligands can be attached to the liposomal surface through various chemical reactions, such as amine coupling, thiol-disulfide exchange, or carbodiimide chemistry (refer to Table 2). Care must be taken to avoid disruption of liposomes during the ligation procedure.

Non-covalent incorporation: ligands can sometimes be placed on the surface of liposomes via non-covalent interactions, such as electrostatic interactions and hydrophobic interactions.

Click chemistry: This refers to a class of high-yield bioorthogonal reactions (i.e., chemical reactions that can occur inside living systems without interfering with the native biochemical processes and functions) that can be used to conjugate ligands onto liposome surfaces. Examples of click chemistry reactions include azide-alkyne cycloaddition (CuAAC) and strain-promoted azide–alkyne cycloaddition (SPAAC). 

Streptavidin–biotin interaction: biotinylated lipids can be incorporated into liposomes, and ligands that are conjugated with streptavidin can be attached to the liposomal surface via the strong and specific interaction between streptavidin and biotin.

The choice of conjugation method depends on various factors such as the nature of the ligand, the stability requirements, the desired density of ligands on the liposomal surface, and the intended application of the ligand-modified liposomes [45,46]. It is important to consider the compatibility of the conjugation method with the liposomal formulation and to optimize the conditions to ensure the stability and functionality of the liposomes [47].

**Table 2 ijms-24-13262-t002:** Comparison of different post-insertion moiety conjugation methods [30,45,46,48,49,50,51,52].

Conjugation Method	Description	Schematic	Advantages	Disadvantages
Crosslinking of primary amines by glutaraldehyde	Dimethyl suberimidate or glutaraldehyde can be employed for imidoester or imine crosslinking, respectively. This involves reacting these substances with both the amine-functionalized liposome and the targeting ligand, which possesses an amine group as well.	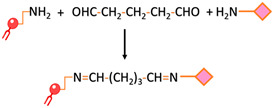	-Preserves the affinity, specificity, and targeting -Common approach	-Homobifunctionality of the linkers -Side reactions -Loss of targeting ligand
Carbonyl-amine bond	Coupling the N-terminus of the peptide to preformed carboxy-terminated liposomes that contain DSPE-PEG involves amine-carboxyl coupling.	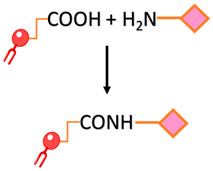	-Straightforward preparation	-The imine bond formed between the carbonyl group of the ligand and the amine group of the target molecule is not stable and can undergo hydrolysis, leading to bond cleavage. -pH sensitivity -The kinetics of the carbonyl amine reaction can be relatively slow, requiring longer reaction times and elevated temperatures to achieve efficient conjugation.
Cyanuric chloride	Nucleophilic substitution takes place under alkaline conditions. The initial two chloride substitutions can be accomplished by reacting with nucleophiles under mildly basic conditions. The first substitution on cyanuric chloride happens rapidly at 0 °C, while the second substitution takes place within 12–24 h at room temperature. The third substitution typically occurs within 12–24 h as well, but it necessitates temperatures above 60 °C.	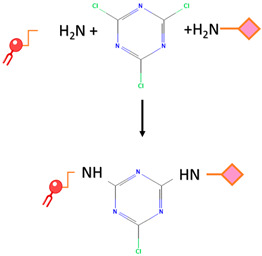	-The safety of cyanuric chloride can be attributed to its pH-regulated reactivity.	-Cyanuric chloride has low reactivity under neutral physiological pH
Amide bond	An amide bond is formed through the reaction of para-nitrophenylcarbonyl with primary amine	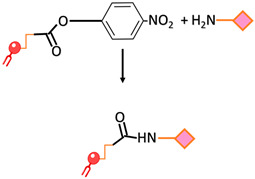	-Undesirable side reactions are circumvented by the hydrolysis of unreacted pNP molecules. -No activation of the carboxylic acid is required simplifying the reaction and decreasing the overall time required for the surface modification	-The pNP lipid must be maintained at acidic pH during its manipulation to prevent the hydrolysis of the pNP group
Disulfide bond	The conjugation of two thiol functions to form a disulfide bond	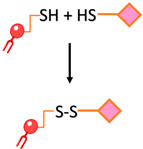	-The reaction can be monitored spectroscopically by the release of the chromophore 2-thiopyridone -Rapid and easy	-Disulfide bonds are relatively unstable under the reductive conditions in serum
Maleimide bond	Thioester bond formation by the maleimide-thiol addition reaction. This process involves the rapid reaction of thiolated ligands with maleimide-modified PEGylated phospholipids such as maleimide-polyethyleneglycol	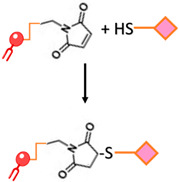	-Most frequently used approach to conjugate targeting ligands onto PEGylated lipids.	-Hydrolytic instability -Sensitivity to reducing agents -Variable reaction kinetics
EDC/NHS	The carboxyl group of a targeting ligand is activated by 1-ethyl-3-(3-dimethylamino-propyl)carbodiimide (EDC) and N-hydroxysuccinimide (NHS) and rapidly reacts with phospholipid-PEG-NH_2_ via nucleophilic substitution reaction to form a stable amide bond.	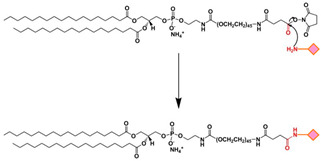	-High specificity -Mild reaction conditions -Compatible with a wide range of moieties	-Generally, EDC/NHS coupling reaction occurs faster in a weak acid solution.

## 2. Transferrin-Liposomes in Cancer Therapy

The hallmarks of cancer include uncontrolled cell division, evasion of growth suppressors, metastasis, and the ability to stimulate the growth of new blood vessels (angiogenesis) [53,54,55,56]. Cancer cells require more iron than normal cells to support their rapid growth rate and proliferation. This high demand for iron results in the overexpression of TfRs in tumor cells. Iron plays a complex role in cancer, and its importance can vary depending on the specific type of cancer and its stage [29,30,38]. Iron is an essential nutrient crucial for various cellular processes, including cell growth and proliferation. Cancer cells often have a high demand for iron to support their rapid growth and division. Iron is also necessary for the synthesis of DNA, which is essential for cell replication. Cancer cells require abundant DNA synthesis to sustain their uncontrolled growth [57,58,59]. Moreover, tumors can trigger angiogenesis (i.e., the formation of new blood vessels), which is essential for tumor growth and metastasis [60]. Iron is involved in angiogenesis through its role in the production of a vascular endothelial growth factor (VEGF), a protein that promotes blood vessel formation. Increased iron levels can stimulate angiogenesis, facilitating tumor growth [61,62,63]. Iron can participate in the generation of reactive oxygen species (ROS), which are highly reactive molecules that can cause damage to cellular components. Cancer cells often exhibit increased levels of oxidative stress, and iron can contribute to this process. ROS can promote genetic mutations and genomic instability, potentially accelerating cancer development (refer to Figure 4) [59,64,65,66].

It is important to note that the relationship between iron and cancer is complex and multifaceted, and research is ongoing to fully understand its implications. The role of iron in cancer development and progression can vary depending on the specific context, and it is a subject of further investigation in the field of cancer research. Due to the overexpression of TfRs on different cancer cells and their specific pathway characteristics, TfRs have emerged as a promising therapeutic approach for treating cancer. Surface conjugation with Tf molecules has been extensively studied and accounts for approximately 43% of the strategies utilized [31].

## 3. Tf-Targeted Liposomes for Glioblastoma Therapy

Glioblastoma multiforme (GBM) is the most aggressive type of glioma, which is a group of tumors that originate from glial cells or their precursors in the central nervous system (CNS). Gliomas account for 80% of malignant brain tumors [67,68]. The survival rate of patients diagnosed with malignant gliomas, for both adults and children, is only 34.4% [29]. Gliomas are clinically classified into four grades, with grade 4 or GBM being the most common and aggressive in humans. GBMs pose significant challenges for therapeutic interventions due to their complex nature. Macroscopically, GBM exhibits a multiform appearance with regions of necrosis and hemorrhage. Microscopically, it displays characteristics such as pseudopalisading necrosis, pleomorphism, and microvascular proliferation [68]. Genetically, GBM demonstrates various deletions, amplifications, point mutations, and intratumor genetic heterogeneity [68,69,70,71]. Primary GBMs (i.e., GBMs that develop without any clinical or histologic evidence of precursor lesions) are usually characterized by three main genetic alterations, namely, EGFR amplification, CDKN2A-p16^INK4a^ gene, and PTEN gene deletion [72,73,74,75]. Table 3 and Figure 5 summarize the aforementioned and other additional genetic alterations associated with primary versus secondary GBM. These factors contribute to its resistance to treatments. Treating GBM surgically is complicated by its diffuse nature and variability in tumor location. The standard approach involves surgically removing as much of the tumor as is possible to do safely, followed by chemotherapy, typically designed to damage DNA or inhibit DNA replication [76,77]. However, the presence of the blood–brain barrier (BBB) adds an additional challenge to GBM treatment [71,78]. The BBB is a highly selective protective barrier characterized by tight extracellular junctions and efflux transporters that prevent the entry of harmful substances into the brain. Unfortunately, this barrier also limits the penetration of certain chemotherapeutic agents that could be effective in treating GBM [29,79]. Therefore, it is crucial to devise an innovative approach for the efficient transportation of drugs to the brain. 

One increasingly prominent strategy involves the use of targeted nanoparticles, which utilize ligands specific to receptors for receptor-mediated uptake [80,81]. Among these receptors, the transferrin receptor holds significance as it exhibits high expression in the brain and is able to cross the BBB, making it a promising candidate for targeted nanoparticle delivery systems in gliomas [31,82,83]. 

Lam et al. [84] conducted a study in which Tf-functionalized liposomes were co-loaded with temozolomide and the bromodomain inhibitor JQ1 to treat GBM. The Tf-modified liposomes were observed to accumulate in the endothelial walls of the brain microvasculature, facilitate the delivery across the BBB via receptor-mediated transcytosis, and enhance the effectiveness of drug delivery to the tumor location. These findings indicate that liposomes functionalized with Tf hold great potential for the treatment of glioma patients. Furthermore, when mice were treated with drug-loaded Tf-liposomes, the brain tumors exhibited elevated indicators of DNA damage and apoptosis, resulting in a notable decrease (1.5-to-2-fold) in tumor burden. 

In another study, Jhaveri et al. [85] encapsulated the drug Resveratrol (RES) in Tf-PEG-PLA liposomes. The developed liposomal formulations were shown to successfully protect the drug, increase its circulation time and improve its stability. Flow cytometry and confocal microscopy were used to investigate the internalization of Tf-conjugated liposomes (Tf-RES-Ls) in U-87 MG cells. Compared to free RES and RES-loaded liposomes (RES-Ls), Tf-RES-Ls exhibited significantly higher cytotoxicity and induced elevated levels of apoptosis in the GBM cells. The cytotoxic effects of the RES were attributed to its ability to halt cell progression in the S-phase of the cell cycle and selectively induce a generation of reactive oxygen species (ROS) in the cancer cells. Xenograft mice were used to evaluate the therapeutic efficacy of the developed RES liposomal formulations in vivo, as well as to conduct a tumor growth inhibition study and a survival rate study. The results demonstrated that Tf-RES-Ls surpassed other treatments in their effectiveness in inhibiting tumor growth and improving overall survival in mice. 

Li et al. [86] prepared Tf-modified liposomes encapsulating elemene (ELE) and cabazitaxel (CTX). The liposomes were further modified with cell membrane proteins of RG2 glioma cells (active-targeting biomimetic liposomes, Tf-ELE/CTX@BLIP). The in vitro experiments demonstrated significant targeting capabilities and immune evasion compared to traditional liposomes (ELE/CTX@LIP), with a 5.83-fold higher intake rate. Bioluminescence imaging of the orthotopic glioma nude mouse model revealed an enhanced drug accumulation in the brain and improved the tumor penetration of Tf-ELE/CTX@BLIP. In vivo, studies showed that the treatment group injected with Tf-ELE/CTX@BLIP showed an increased survival time and a decreased tumor volume. The efficacy of Tf-ELE/CTX@BLIP was indicated by a decrease in the average tumor fluorescence intensity following intravenous administration (65.2, 12.5, 22.1, 6.6, 2.6, and 1.5-times less) compared to the control, CTX solution, ELE solution, ELE/CTX@LIP, ELE/CTX@BLIP, and Tf-ELE/CTX@LIP groups, respectively. Histopathological analysis indicated that Tf-ELE/CTX@BLIP exhibited reduced toxicity compared to the administration of CTX solution. 

In another study, Lakkadwala et al. [87] conducted a thorough investigation into the effectiveness of liposomes that were loaded with two drugs (Doxorubicin and Erlotinib) and functionalized with Tf and a cell-penetrating peptide called penetratin (Pen) at targeting GBM in a mouse model. The results showed that the Tf-Pen liposomes exhibited around a 12- and 3.3-fold increase in doxorubicin (DOX) and erlotinib accumulation in mice, respectively, compared to free drugs. Furthermore, the use of Tf-Pen liposomes demonstrated remarkable effectiveness in inhibiting tumor growth, leading to a 90% reduction in tumor size in the mice. Additionally, this treatment approach resulted in a substantial increase in the median survival time (36 days), without any observed toxicity. Table 4 presents a summary of some studies focused on Tf-functionalized liposomes for GBM therapy.

## 4. Challenges and Future Directions

Despite the promising results reported on Tf-conjugated liposomes in GBM therapy, some challenges need to be addressed. For instance, many NPs-based experimental approaches do not reach clinical evaluation due to the complications introduced by the BBB [90]. There are a few clinical trials using NPs that have the potential to be applied for GBM treatment (ClinicalTrials.gov Identifiers: NCT00355888, NCT00944801, NCT00734682, NCT05768919, NCT01906385, NCT03086616). Table 5 summarizes some clinical trials that are studying the use of liposomes for GBM therapy. However, further research is still needed to enhance the therapeutic potential of these strategies for GBM therapy. Some of the aspects that challenge the clinical translation of nanotherapeutics include their possible toxicity, reproducibility and batch-to-batch variability in large-scale production and targeting ability [90,91].

With regard to toxicity, all the components making up a drug delivery system should be non-toxic and approved by the proper authorities [92]. This is nearly always the case for liposomes.

As nanotechnology advances, a wide range of new materials is being developed for potential use in drug nanocarriers. However, obtaining clinical approval for these new materials can be a time-consuming process. As a result, researchers tend to concentrate on using materials that are already known and approved, which hinders the translation of novel materials into clinical practice [93,94]. The large-scale production of nanotherapeutics is challenged, in part, by reproducibility issues. Although there are numerous protocols available for synthesizing NPs for various therapies, including those targeting GBMs, these protocols do not always yield reproducible results. Consequently, further modifications and adaptations of published protocols become necessary for each specific case [90,95,96]. Additionally, stringent quality control requirements must be met, depending on the type of drug carriers employed [97]. 

With respect to in vivo targeting ability, when NPs are administered systemically to transport substances across the BBB, there is a possibility that they may become coated with serum albumin, immunoglobulins, and other blood components. This coating can cause the NPs to be engulfed by macrophages and results in a substantial accumulation of NPs in the liver and spleen. Consequently, only a small portion of the drugs actually reach the brain tumors, leading to a significant reduction in the effectiveness of the therapy [90,98]. Coating NPs with stealth-imparting polymers such as polyethylene glycol (PEG) has been proposed to address this issue because it can hide the NPs from the immune system and increase their circulation time in the organism [31,99,100]. However, PEG has been associated with an unanticipated immune response known as the accelerated blood clearance (ABC) phenomenon [101,102]. The ABC phenomenon has been widely documented in cases involving the repetitive use of PEGylated substances as well as PEGylated nanocarriers, where the presence of PEG increases their elimination rates and diminishes their effectiveness [103,104,105,106]. Moreover, functionalizing the surface of NPs with targeting ligands can result in the formation of a protein corona. This protein corona has a significant impact on the properties of the NPs, influencing their behavior in vivo and disrupting the ligand-receptor interactions leading to a loss of targeting capability. Additionally, it can hinder lysosomal escape and the transcytosis process across the BBB [107,108,109].

Various strategies have been proposed to mitigate the nonspecific adsorption of the protein corona. These include surface modification with stealth polymers such as PEG and the use of biomimetic membrane camouflage. Another intriguing approach involves pre-coating the NPs with specific proteins or peptides to regulate the formation of the protein corona. For instance, iron-mimicking peptides such as T10 and CTR can be used to facilitate the recruitment of the Tf corona onto NPs in situ. Upon systemic administration, these peptides bind to the naturally occurring Tf present in the bloodstream, forming a corona around the NPs’ surface. This in turn enables the selective transportation of NPs across the BBB and their internalization into the targeted tumor cells. Since these peptides attach to the nonbinding regions of the Tf molecule, this binding event does not disrupt the biological function of endogenous Tf [31,90,110].

Intranasal (IN) delivery has been extensively researched as a method of delivering drugs for various medical conditions. Recently, there has been a particular focus on using IN delivery to bypass the BBB and directly target the brain, and treat central nervous system (CNS) disorders. The nasal route offers several advantages for drug delivery because it is easily accessible, highly vascularized, and allows for lower medication doses because it can circumvent hepatic metabolism; the aforementioned advantages make IN particularly relevant in GBM therapy. The irritation to the nasal cavity and mucociliary clearance associated with IN delivery can be overcome by using nanocarriers, which are biocompatible formulations and are easily absorbed by the nasal mucosa [111,112]. Semyachkina-Glushkovskaya et al. [113] conducted a study investigating the IN delivery of monosialotetrahexosylganglioside (ganglioside GM1)-loaded liposomes accompanied by the photostimulation of the lymphatic vessels using near-infrared light (NIR). The results showed that the IN-delivered nanocarriers reached the tumor site through the lymphatic system and that the photostimulation improved the anti-cancer effects by generating reactive oxygen species (ROS). 

Another method proposed to facilitate the transport of chemotherapeutic agents across the BBB involves the disruption of the BBB by using focused ultrasound (FUS) [31,90]. Research has indicated that combining low-intensity focused ultrasound (LIFU) with microbubbles can lead to the localized and temporary disruption of the BBB [114]. Accordingly, Lin et al. [115] utilized DOX-loaded cationic liposomes along with LIFU to penetrate the BBB and target C6 glioma in a rat model. The use of LIFU facilitated BBB opening, enabling liposomes to deliver DOX to the glioma. This combinational approach prolonged glioma inhibition with minimal side effects. In another study, Papachristodoulou et al. [116] demonstrated that LIFU can effectively deliver liposomes to the tumor region, as evidenced in mice with TMZ-resistant gliomas.

Moreover, Tf as a targeting moiety against the TfR suffers from certain limitations such as loss of specificity due to abundance of endogenous Tf. The presence of plasma Tf molecules can compete with Tf-conjugated NPs for binding to the TfR, leading to a decrease in targeting specificity and therapeutic efficacy. Furthermore, the TfR is also expressed on the surfaces of normal cells; therefore, Tf-conjugated liposomes may enhance the off-target uptake of chemotherapeutics and cytotoxicity to these normal cells [117]. The off-target accumulation of these Tf-conjugated liposomes in clearance organs, such as the spleen, could potentially trigger an innate immune response, whereas an accumulation in the liver has the potential to serve as a storage site for prolonged release. This could significantly impact the overall dosage and frequency of administration throughout the body [118,119]. To overcome these drawbacks, alternative strategies have been investigated including ligands that bind to the receptor at non-competitive sites. These alternative ligands, such as monoclonal antibodies (mAbs), do not compete with the circulating Tf, avoiding TfR saturation and enhancing specificity. However, the synthesis of mAbs is challenging especially when it comes to controlling their quality, which limits their clinical application. Additionally, the large molecular weight of antibodies and Tf molecules hinders their effective use, highlighting the need for more efficient targeting strategies [31,90,120]. 

In recent years, single chain fragment variables (scFvs) have emerged as an alternative to antibodies to target the TfR. These scFvs retain the binding specificity of antibodies while demonstrating improved pharmacokinetic properties. Their smaller size allows for better tissue penetration and faster clearance from blood and tissues. Although antibodies and their fragments offer several advantages, their use in clinics would result in a substantial rise in treatment expenses. As a result, there have been extensive efforts to explore more economical alternatives [31,90,120,121]. Consequently, TfR-targeting peptides have gained popularity as an alternative to antibodies because although these peptides bind to TfRs without competing with plasma Tf, their smaller size gives them a significant advantage over regular antibodies and/or Tf molecules in terms of transportation across the BBB [120,122]. Studies have shown that higher molecular weight ligands tend to obstruct the transportation of the actively targeted NPs across the BBB. Hence, surface modification with the smaller-sized TfR-targeting peptides enables the maintenance of binding efficiency, and high receptor specificity [31,87,90,123].

## 5. Conclusions

Recently, NPs have emerged as a promising therapeutic alternative capable of overcoming the shortcomings of chemotherapy, mainly poor biodistribution and significant toxicity to healthy tissues. However, passive targeting strategies based solely on NPs’ physicochemical properties have proven ineffective due to the lack of tissue specificity. To address this issue, active targeting strategies based on the overexpression of the TfR in brain tissue were developed. Currently, targeting the TfR remains one of the most popular approaches for enhancing the delivery of anti-cancer therapeutics to GBM. Despite scientific advancements, the progress in clinical trials has been limited primarily due to the high costs and complex production methods associated with NP-based therapeutics. Further research is necessary to optimize TfR targeting and to explore the full potential of TfR-targeting liposomes for drug delivery in GBM. Such advancements would significantly impact cancer research and ultimately contribute to improving the quality of life and survival chances of GBM patients.

## Figures and Tables

**Figure 1 ijms-24-13262-f001:**
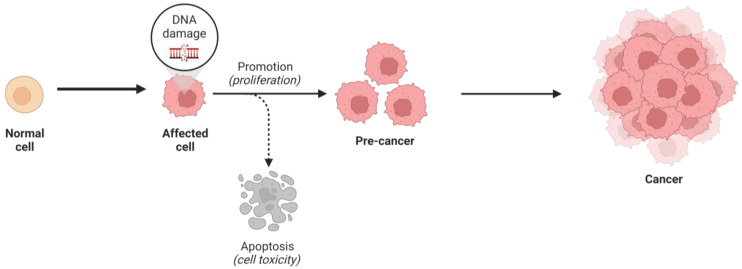
The process of tumorigenesis (created using BioRender.com accessed on 16 January 2023).

**Figure 2 ijms-24-13262-f002:**
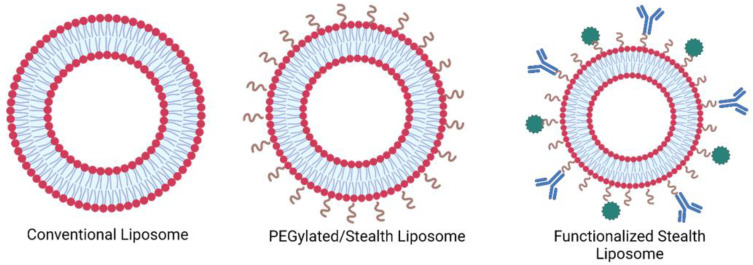
Surface functionalization of liposomes (created using BioRender.com accessed on 30 May 2023).

**Figure 3 ijms-24-13262-f003:**
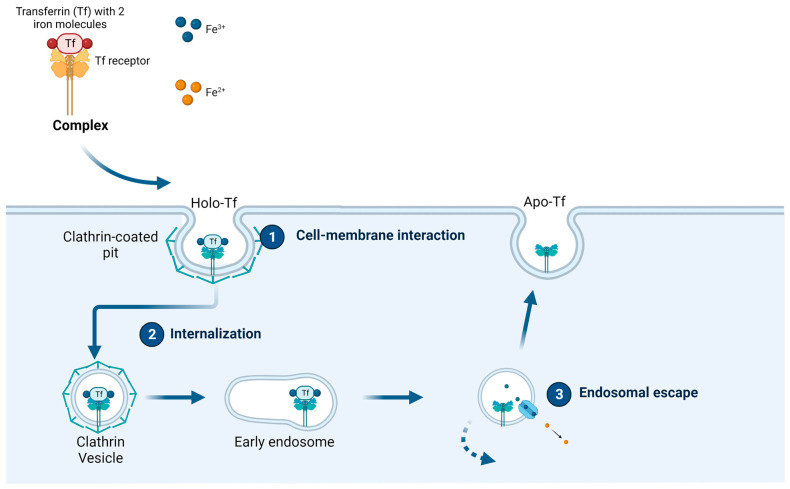
Clathrin-mediated endocytosis (created using BioRender.com accessed on 22 May 2023).

**Figure 4 ijms-24-13262-f004:**
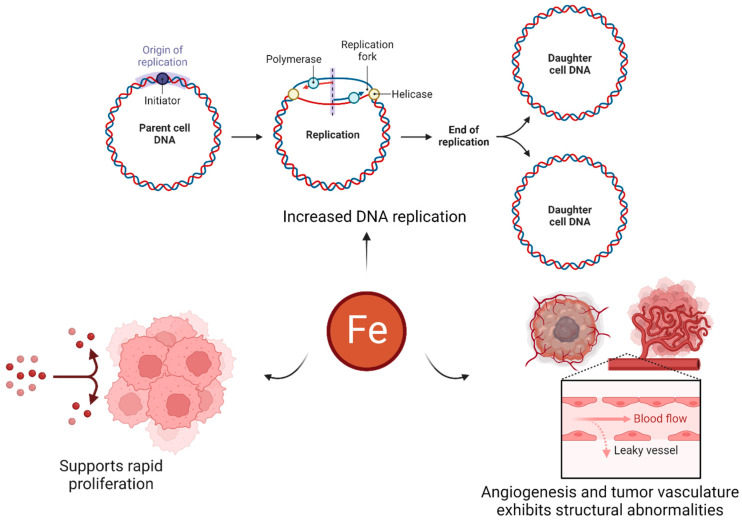
Role of iron in cancer (created using BioRender.com accessed on 6 June 2023).

**Figure 5 ijms-24-13262-f005:**
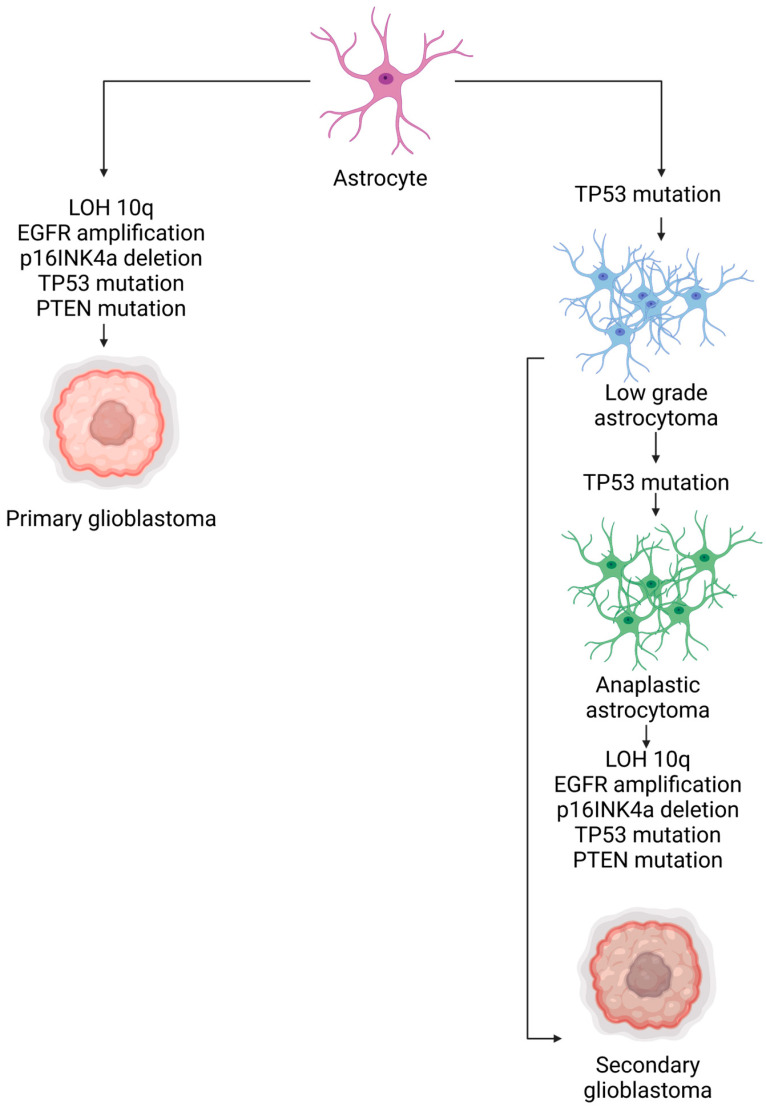
Genetic mutations of GBM (created using BioRender.com accessed on 8 July 2023).

**Table 1 ijms-24-13262-t001:** List of some clinically approved liposomal chemotherapeutics [20,21].

Trade Name	Year Approved	Active Ingredient	Use
Doxil/Caelyx	1995	Doxorubicin	Antineoplastic
Daunoxome	1996	Daunorubicin	Antineoplastic
DepoCyte	1999	Cytarabine	Lymphomatous meningitis
Myocet	2001	Doxorubicin	Metastatic breast cancer
Mepact	2009	Mifamurtide	Osteosarcoma
Marqibo^®^ (Onco TCS)	2012	Vincristine	Acute lymphoblastic leukemia

**Table 3 ijms-24-13262-t003:** Gene mutations common in GBM (adapted from [72,73,74,75]).

Gene Name	Abbreviation	Function of Encoded Protein
Epidermal growth factor receptor	EGFR	Regulating cell proliferation and survival
Isocitrate dehydrogenase 1	IDH1	Production of NADPH
Neurofibromin 1	NF1	Regulating cell proliferation and survival
Phosphoinositide-3-kinase catalytic alpha	PIK3CA	Regulating cell proliferation and survival
Phosphoinositide-3-kinase regulatory 1	PIK3R1	Regulating cell proliferation and survival
Phosphatase and tensin homolog	PTEN	Regulating cell proliferation and survival
Protein tyrosine phosphatase receptor type D	PTPRD	Regulating cell proliferation and survival
Retinoblastoma 1	RB1	Regulating cell cycle
Tumor protein p53	TP53	Apoptosis
V-erb-b2-erythroblastic leukemia viral oncogene homolog 2	ERBB2	Regulating cell proliferation and survival

**Table 4 ijms-24-13262-t004:** Studies on Tf-targeted liposomes for GBM therapy.

Payload	Tf conjugation Method	Glioma Cell Line Used	Animal Model	Main Findings	Ref.
Cisplatin	Coupling the terminal carboxy group of DSPE-PEG2000-COOH	bEnd3/C6 co-culture	-	-Tf-modified liposomes improved transport across BBB using clathrin-mediated endocytosis -Tf-modified liposomes showed higher toxicity to glioma cells in vitro	[88]
Temozolomide, bromodomain inhibitor JQ1	Post insertion	U87MG, GL261	NCR nude mice	-Tf-modified liposomes improved delivery through receptor-mediated transcytosis -Mice in Tf-modified liposomes treatment group showed increased markers for DNA damage and apoptosis -1.5-to-2-fold decrease in tumor burden in mice	[84]
Resveratrol	Post insertion	U-87 MG	Female athymic NCr-nu/nu nude mice	-There was a 2.3-fold higher association between Tf-modified liposomes and glioma cells compared to normal cells expressing TfRs -The percent cell viability for Tf-RES-liposomes was 54.2 ± 3.8% while that for RES-liposomes was 61.9 ± 3.9% -Mice in the Tf-modified liposomes had the longest survival rate of 28 days.	[85]
Elemene, Cabazitaxel	-	U251, C6, RG2	Female nude mice	-In vitro, studies showed a 5.83-fold higher intake rate of Tf-modified liposomes -Tf-modified liposomes increased the mean survival time from 28 days (for non-targeted liposomes) to 30 days	[86]
Doxorubicin, erlotinib	Tf was coupled to the terminal end of DSPE-PEG(2000)-NHS using nucleophilic substitution	U87, bEnd.3	Male/female nude mice	-The transport of the drugs by Tf-Pen-modified liposomes was improved by 15% in vitro -Tf-Pen liposomes achieved tumor regression in mice brains by ~90% -Tf-Pen liposomes achieved the highest median survival time of 36 days	[87]
Doxorubicin	Post insertion	U87, GL261	Balb/c nude mice	-The MTT assay showed that the antitumor activity of Tf-liposomes was higher than that of non-targeted liposomes and free DOX -There was not much difference in the survival time of mice treated with Tf-liposomes and non-targeted liposomes (25 and 24 days, respectively)	[89]

**Table 5 ijms-24-13262-t005:** Summary of clinical trials testing liposomal formulations for GBM therapy.

Name of Institution	Liposomal Payload	Number of Patients	Description	Status	Identifier (Clinicaltrials.gov accessed on 10 July 2023)
University of Regensburg	Doxorubicin, Temozolomide	63	-Pegylated liposomal doxorubicin is combined with temozolomide and radiotherapy -In phase I: pegylated liposomal doxorubicin increased in increments of 5 mg/m^2^ (starting from 5 to 20 mg/m^2^) -In phase II: the pegylated liposomal doxorubicin targeted dose of 20 mg/m^2^ is administered up to a cumulative dose of 550 mg/m^2^ or until tumor progression	Completed	NCT00944801
Medical University of South Carolina	cytarabine (DepoCyt), Temozolomide	12	-Phase I and phase II intraventricular liposomal cytarabine (DepoCyt) in combination with oral temozolomide	Terminated	NCT01044966
University of California, San Francisco	CPT-11	34	-Phase I dose escalation scheme with a starting dose of 120 mg/m^2^ or 60 mg/m^2^ through IV	Completed	NCT00734682
SignPath Pharma, Inc.	Curcumin	30	-Phase I and phase II dose escalating scheme of liposomal curcumin concurrent with temozolomide and radiation therapy as well as adjuvant temozolomide after radiation therapy	Recruiting	NCT05768919
Plus Therapeutics	186Rhenium	30	-Phase I to determine the maximum tolerated dose of nanoliposomal 186Rhenium -Phase II to determine overall survival	Recruiting	NCT01906385
Emory University	Verteporfin	24	-Phase I and phase II dose escalation study to study the side effects and determine the optimal dose of visudyne (liposomal verteporfin)	Recruiting	NCT04590664
University of Florida	RNA	28	-Phase I study to determine manufacturing feasibility, safety, and the maximum tolerated dose -The trial consists of three parts, surgery, radiation, and immunotherapy -Liposomal RNA vaccine will be administered monthly for a total of 15 vaccines after radiation	Recruiting	NCT04573140

## Data Availability

Not applicable.

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
