# Peer review of "Transferrin-Targeted Liposomes in Glioblastoma Therapy: A Review"

_ijms, 2023, doi:10.3390/ijms241713262_

Round 1

Reviewer 1 Report

This review summarizes the use of transferrin ligand for targeting liposomes to glioblastoma. 

Although the text is comprehensive and you can read it easily, I can't find the innovation of it. With a quick search on pubmed, I've found that there is a very similar review already published: "Transferrin receptor-mediated liposomal drug delivery: recent trends in targeted therapy of cancer" and thus, I think that this review it is not suitable for publication in this journal.

Moreover, there are same mistakes, including the last sentence of the abstract that should be corrected, as well as the quality of the images.

Author Response

Please find our rebuttal attached.

Reviewer 2 Report

Kawak et al., provides a concise overview of transferrin-targeted liposomes in the therapy of glioblastoma. I liked the progression of the review starting from what liposomes are to transferrin (Tf) and its receptors (Tfs), conjugation chemistry of Tf and liposomes, Tf-liposomes in cancer therapy in general followed by Tf-targeted liposomes in brain cancer and challenges and future direction, a format that I think a reasonable review should be.

I have the following comments:

1.       Many reference sources as indicated in bold letters in the manuscript needs to be fixed.

2.       There needs to be a reference at the end of the last sentence on top of page 4 under ‘liposomes’.

3.       There should be a reference on page 14 (line 190?  Note line numbers have not been provided appropriately).

4.       Note that even though the main theme of the review is glioblastoma, only about 25% of the review has been devoted to this theme.

I suggest the authors prepare two more Tables like Table 2.

One Table will have columns heading Conjugation method, description, model used, clinical parameters used, and outcome – this Table will summarize the observations on Tf-targeted liposomes for brain cancer therapy.

The other Table will summarize the results of the clinical trials. This Table will have two more columns (name of Institution and number of patients) in addition to the 5 columns indicated above.

5.       Since TfR is expressed in many cell types, and since Tf-targeted liposomes can bind to these receptors, such ‘unwanted binding’ of the engineered liposomes can be a potential factor underlying toxicity.  I would like the authors to discuss this avenue in the article.

6.       Also discuss potential nasal delivery of the liposomes, especially in the context of the potential of reduced toxicity.

7.       Under ‘Acknowledgements’, the last line says, “The author M.N.Z dedicates this work……”. I think this author should be N.M.A.S, not M.N.Z.  Also, the word ‘compassion’ may be replaced with ‘provided encouragement to pursue science’.

Author Response

Please find our responses attached.

Reviewer 3 Report

The review article by Dr Kawak et al., entitled “Transferrin-targeted Liposomes in Glioblastoma Therapy: A Review”, deals with new therapeutic strategies in GBM, based on the use of transferrin-conjugated liposomes to deliver specific drugs and suppress tumorigenic GBM cells within the brain.

The review describes very appropriately the technical procedure of conjugation of liposomes with transferrin and the main therapeutic approaches which use this method in GBM therapy. However, the review lacks of specific introduction about the biology of GBM and the alterations, which characterize this aggressive and lethal brain tumor. In detail, the points which should be addressed are the following:

-The Authors should refer more specifically to alterations which characterize tumorigenic GBM cells. There is a wide literature, which documents the variety of genetic changes associated with GBM, and this would be reported in a table and schematized in a Figure.

-There are some sentences throughout the manuscript which refer to not found reference (i.e., at the end of page 2). Please add the appropriate reference or delete the sentence for which there is not any supporting reference.

English language requires an extensive revision.

Author Response

Please find our response attached.
